# Acute Effects of Static Self-Stretching Exercises and Foam Roller Self-Massaging on the Trunk Range of Motions and Strength of the Trunk Extensors

**DOI:** 10.3390/sports9120159

**Published:** 2021-11-26

**Authors:** Maria Fonta, Elias Tsepis, Konstantinos Fousekis, Dimitris Mandalidis

**Affiliations:** 1Department of Physiotherapy, School of Health Rehabilitation Sciences, University of Patras, 25100 Aigio, Greece; mariafo1980@gmail.com (M.F.); tsepis@upatras.gr (E.T.); kfousekis@upatras.gr (K.F.); 2Sports Physical Therapy Laboratory, Department of Physical Education and Sports Science, School of Physical Education and Sports Science, National and Kapodistrian University of Athens, 17237 Athens, Greece

**Keywords:** single-bout interventions, flexibility, self-myofascial release, posterior torso, thoracolumbar spine movements, isometric strength

## Abstract

Although the effectiveness of static self-stretching exercises (SSSEs) and foam roller self-massaging (FRSM) in joint range of motion and muscle strength of the lower limbs has been extensively investigated, little is known about their effectiveness on the posterior trunk muscles. The present study aimed to investigate the acute effects of two 7-min SSSEs and FRSM intervention protocols on the range of trunk movements and the strength of the trunk extensors. Twenty-five healthy active males (*n* = 14) and females (*n* = 11) performed each intervention separately, one week apart. The range of motion (ROM) of the trunk-hip flexion (T-HF), the ROM of the trunk side-flexion (TSF) and rotation (TR) bilaterally, as well as the isometric maximum strength (TESmax) and endurance (TESend) of the trunk extensors were measured before and after each intervention. The ROMs of T-HF, TSF, and TR were significantly increased following both SSSEs and FRSM. The TESmax and TESend were also significantly increased after FRSM, but decreased following SSSEs. While both interventions were effective in increasing the range of motion of the trunk, a single 7-min session of FRSM presented more advantages over a similar duration SSSEs protocol due to the increase in the strength of the trunk extensors it induced.

## 1. Introduction

Maintaining soft tissue elasticity and full and unrestricted joint range of motion (ROM) are necessary elements for optimal movement of the whole body, as well as its individual parts, in both daily and sporting activities. Static self-stretching exercises (SSSEs), probably the most common form of stretching exercises, have been the main intervention used for many years by most athletes before and after sports participation to: (i) ensure sufficient joint ROM to perform optimally an athletic activity [1,2]; (ii) decrease the risk for injury by reducing the muscle stiffness or increasing the muscle compliance [3,4]; and (iii) accelerate recovery [5]. However, whilst most studies have shown that a single bout of SSSEs may lead to an immediate increase of joint mobility, under certain conditions it may have detrimental effects on sports performance [1,2,6]. This is due to the time-dependent effect of static stretching, as the stretches that last longer periods (≥60 s) cause an immediate and relatively greater reduction in strength, speed, and power tasks than the ones that last less (<60 s) [1,2,6]. To avoid the potential negative effects of static stretching, many athletes have sought other methods to maintain muscle elasticity and preserve the joints’ ROM. Among the most popular interventions that they implemented are the self-assisted myofascial-release techniques using equipment such as foam or massage rollers [7,8]. Essentially, these techniques are forms of soft tissue self-massaging that aim to release the soft tissue from the pathological traction exerted by a fascia that has become either inelastic or adherent to adjacent tissues due to injury or pathology [9,10]. Although it is not clear whether these techniques release myofascia, [8], their effectiveness is clinically manifested with an increase in soft tissue elasticity and joint ROM, but unlike static stretching, they seem to prevent a decrease [11,12,13,14,15] or even improve muscle strength [16,17].

The effectiveness of a single bout of SSSEs over self-massaging techniques using either a foam or a massage roller, and vice versa, on the ROM and/or muscle strength has been investigated mainly on the lower limb musculature. Halperin et al. found that three sets of 30 seconds with 10 s of rest between the sets of both massaging with a roller and static stretching increased ankle passive dorsiflexion ROM by approximately 4% and 5.2%, respectively, at 1 and 10 min after the interventions in 14 recreationally trained subjects [14]. However, only the massage roller demonstrated small improvements in maximum voluntary contraction (MVC) force at 10 min postintervention, being significant different compared to static stretching exercises that incidentally resulted in a decreased maximal force output at the same time point [14]. Su et al. showed that foam rolling was more effective than static and dynamic stretching by acutely increasing the flexibility of quadriceps and hamstrings without hampering muscle strength, and thus it may be recommended as part of a warm-up in healthy young adults [18]. Lee et al. demonstrated an increased knee extension ROM following self-massaging with a foam roller of the hamstrings, as static stretching exercises did, with a concomitant increase in quadricep strength [19]. Static stretching had no effect on the strength of either the quadriceps or the hamstrings. Despite the extensive implementation of SSSEs and foam roller self-massaging (FRSM) in the lower limb, the effectiveness of these relaxation techniques on the trunk musculature is limited, and their use is partly based on empirical data. There is evidence that a single bout of SSSEs, either alone or following a general warming-up bout, increases the range of flexion, side flexion, and axial rotation in both the general [20,21] and the athletic population [22]; however, the use of a foam roller as a muscle relaxant means on the trunk musculature has been poorly investigated, as there are only a few studies that have dealt with this issue so far [23].

The mobility, stability, and neuromuscular control of the trunk required to perform both daily [24,25] and more complex sporting activities [26] are inextricably linked to the normal function of the myofascial tissues of the posterior trunk. Many of the posterior trunk structures, such as those extending the thoracic and lumbar spine, are prone to tightness or shortness and may induce muscle imbalance, leading eventually to movement dysfunction [27]. In addition, these muscles counteract gravity’s pull forwards, enabling a person to maintain an optimum upright posture during relaxed standing or to withstand loads that are applied on their body during demanding sport- and work-related postures and movements (e.g., load transfer). It is therefore necessary for trainees, sports scientists, and healthcare providers (e.g., physical therapists or athletic trainers) to know the technique or method that will enable them to optimize the mobility and muscle strength of the trunk for both preventive and rehabilitative purposes. The present study aimed to investigate the acute effects of a single bout of SSSEs and FRSM on the range of trunk movements and the trunk extensors’ muscle strength and endurance.

## 2. Materials and Methods

### 2.1. Subjects

A total of 25 healthy active individuals, out of 31 who were assessed for eligibility, who performed moderate-intensity exercises at least 2–3 times a week, and including 14 males (age: 31.2 ± 5.2 yrs., height 178.8 ± 6.1 cm, body weight: 84.9 ± 11.3 kg) and 11 females (age: 29.8 ± 5.5 yrs., height: 171.4 ± 4.6 cm, body weight: 60.3 ± 5.2 kg) volunteered for this study (Figure 1). Individuals with pain or discomfort that was present during the last six months or emerged at the beginning or during the study, symptom-free chronic pathologies (e.g., intervertebral disc herniation or spondylolisthesis) or injuries (e.g., spinal fractures) in the neck and trunk, as well as neurological or rheumatologic problems, were excluded from the study. Exclusion criteria also included excessive scoliosis (>5.0° trunk rotation with Adam’s test) and leg length discrepancy more than 0.5 cm, as well as previous experience in using foam rollers. The magnitude of scoliosis was determined based on the magnitude of the rotational deformity (in degrees) that was noted at the thoracic or lumbar spine by placing gently a bubble-level-based scoliometer at the apex of the deformity, perpendicular to the long axis of the body during the “Adam’s forward bend test” [28]. Leg length discrepancy was determined by measuring the distance between the anterior superior iliac spine to the inferior tip of the medial malleolus of each lower limb with a standard measure tape while keeping the ankle, the hip and the subtalar joint neutrally positioned [29]. All subjects signed an informed written consent prior to testing.

### 2.2. Study Procedure

The study was carried out using a crossover experimental design method, with the participants randomly performing two 7-min protocols of SSSEs and FRSM on two separate occasions. Participants were assigned to the first intervention after creating a random series of 25 pairs of interventions (SSSEs and FRSM) balanced in such a way that 13 of the participants would eventually have to perform first the SSSE protocol and the other 12 the FRSM protocol. The series of random pairs was created by the last author using a web application (https://www.random.org accessed on 22 November 2021), and was revealed to the examiner but not to the examinees, who were ultimately blind to the treatment protocols and their possible effects. A washout period of one week between interventions was given to avoid transfer of the effects between techniques (Figure 1). Prior to each intervention, all participants performed a light warm-up program, which included 5 min of running on a treadmill or cycling on a cycle ergometer and 5 min of trunk mobility exercises. The active range of trunk movements was always measured before the strength of trunk extensors, and all tests were performed before and after interventions.

#### 2.2.1. Foam Roller Self-Massaging Protocol

The FRSM protocol, which included two sets of 10 cephalocaudal rolls with each set lasting 30 sec and 15 sec rest between sets, was performed with each participant rolling with the dorsal aspect of the trunk: (i) from the supine position against the floor along the central and lateral surfaces of the thoracic (3 min) and lumbar spine (3 min) using a 30 cm × 15 cm medium-hardness cylindrical foam roller (MED Foam Roller, Blackroll^®^, Bottighofen, Switzerland); and (ii) from the upright standing position against the wall along the paravertebral muscles of the trunk (1 min) using a 24 cm × 12 cm peanut-type foam roller (MED Foam Roller and Duoball Fascia Ball, Blackroll^®^, Bottighofen, Switzerland). Rolls were performed so that the pressure exerted by the participants’ weight caused minimal pain or discomfort. The rate of rolling was set by a metronome at 20 beats per min (Figure 2a–c). The total duration of each session, based on the duration of each set (30 s), the number of sets (*n* = 2), and the scrolling instructions (*n* = 7) was 7 min. Although rolling time did not appear to significantly affect the range of motion [30], most studies have shown that this, as well as performance, may be improved with rolling that lasts up to 300 s per muscle and up to 450 s per session [31]. The program was designed to meet these criteria, considering at the same time the wide area where the muscles of the dorsal surface of the trunk are located and the possible discomfort that could be caused by pressure exerted by the bodyweight of the practitioners on the foam roller. Furthermore, the cylindrical foam roller was selected, as it was expected to potentially affect the more superficial located soft tissues structures that cover the wide area of the dorsal aspect of the trunk (i.e., the thoracolumbar fascia, the latissimus dorsi, the trapezius, rhomboids, and serratus posterior). The peanut-type foam roller, on the other hand, was chosen because, according to the manufacturer’s instructions, it is particularly suitable for all parallel muscle groups, such as the erector spinae (i.e., spinalis and longissimus thoracic, iliocostalis thoracis and lumborum, and multifidus), and it can penetrate deeper into the tissue.

#### 2.2.2. Static Self-Stretching Exercise Protocol

The 7-min SSSE protocol included stretching exercises for: (i) the trunk extensors; (ii) the right and left thoracic; (iii) the lumbar trunk rotators; as well as (iv) the right and left trunk side flexors (Figure 3a–d). Other soft tissue structures, such as the thoracolumbar fascia, the latissimus dorsi, and the trapezius muscle, may have eventually been stretched, as the suggested exercises required additional movements of the upper and lower limb joints. All participants were instructed to perform each exercise twice, with 15 s of rest between repetitions, at a slow speed and towards the right and left side (for the trunk side flexors and rotators only), after the exercise was demonstrated by the primary investigator. To avoid inconsistencies in execution time between the two protocols and to minimize the potential effect of stretching exercises on muscle strength, each participant was instructed to remain in the final position for 30 s while maintaining the intensity of stretching just before the point of discomfort [32].

### 2.3. Outcome Measures

#### 2.3.1. Range of Motion Measurements

The active range of motion (ROM) of right and left trunk side flexion (TSFR and TSFL) and rotation (TRR and TRL) was measured using an inertial measurement device that could supply acceleration (up to 16 g), angular velocity (up to 2000°/s), and magnetic field measurements via a built-in 3D accelerometer, a 3D gyroscope, and a 3D magnetometer, respectively, with an acquisition frequency of 1000 Hz (Gyko, Microgate S.r.l., Bolzano, Italy). A previous study revealed good reliability and excellent concurrent validity of this device in measuring the ranges of trunk movements, supporting its use in clinical practice [33]. The device was attached to a vest, which was worn and adjusted on each participant’s upper torso during testing so that the sensor was located between the two scapulae at the level of the roots of the scapulae spines. The data was transmitted wirelessly in real time to a computer via Bluetooth technology, through which it was possible to display and store it for future use and analysis using computer software (Gyko RePower, Microgate S.r.l., Bolzano, Italy). The active range of trunk side flexion was measured with each participant in the upright standing position while facing and maintaining the palmar surface of their hands in contact with the wall [34]. Each participant was advised to flex their trunk to the right and left sides by sliding their hands on the wall, avoiding forward bending or twisting of the trunk and keeping the elbows and knees straight, as well as the heels on the ground. The active range of trunk rotation was measured in the horizontal level with each participant in the upright sitting position. The effects of potential lower body and pelvic movements on trunk rotation measurements were prevented by instructing each participant to gently pressure a cylindrical cushion that was placed between their knees [35]. Trunk-hip flexion (T-HF) was measured with the Sit-and-Reach Test, a test usually used to assess trunk extensor and hamstring flexibility, with each participant leaning the trunk forward as far as possible from the long sitting position and sliding a metal plate with the fingertips while holding the knees straight and the feet flat against the Modified Sit-and-Reach box (Acuflex^®^ I, Novel Products, Inc., Rocton, IL, USA). An attempt was considered valid and measured if the final position was achieved at slow speed without bouncing and maintained for 2–3 s. Each participant was instructed to perform two trial attempts after each test was demonstrated by the examiner, and then three more attempts, the average of which was used in the data analysis. A 10 s rest was given between the attempts, and 2 min break was allowed between the tests.

#### 2.3.2. Muscle Strength and Endurance Measurements

The trunk extensors’ maximum strength (TESmax) was measured isometrically in Newtons (N), with each participant in the prone position on the examination table [36] using an S-type digital force gauge for tension and compression measurements (measuring range: ≤50 kΝ; acquisition frequency: 2000 Hz; precision: 0.5% of max; FH-5K, Sauter GmbH, Germany). One side of the load cell was connected to a wooden platform that was placed under the front legs of the examination table. The other side was attached to a vest, which was worn by each participant during testing, via a metal ring located on the anterior surface of the chest (midway between the xiphoid process and sternoclavicular groove) using a heavy-duty strap that was passed vertically through an opening at the upper part of the table. The vest was adjusted firmly on each participant’s upper torso using adjustable straps, whilst two more straps were placed around the pelvis and the lower third of the legs to stabilize each participant on the examination table. Each participant was instructed to extend the trunk twice, avoiding abrupt movements and keeping the upper limbs on the side of the body with near maximum effort, and perform three more attempts of 5 s each with maximum effort, the average of which was used in the data analysis. A 1 min rest was given between the attempts (Figure 4a).

The trunk extensors’ muscle endurance (TESend) was also measured isometrically with the Sorensen Test [37]. This test essentially measures the time (in seconds) that a participant can hold their upper torso (from the level of the anterior iliac spine and above) at a horizontal position outside the examination table with the upper limbs crossed in front of the chest. The test was performed with each participant in the prone position on the examination table on which they were stabilized with straps around the pelvis and the lower third of the legs. An antenna, which was mounted horizontally on a stadiometer, was adjusted in height and length to provide the participants with tactile stimuli on the dorsal surface of their trunk regarding its horizontal position. The test was completed when a participant could no longer maintain their upper torso in contact with the antenna for ≥10 s (Figure 4b).

#### 2.3.3. Reliability of the Outcome Measures

The reliability of the outcome measures was determined prior to the commencement of the study based on findings that were published in previous studies. The intraclass correlation coefficient (ICC), the standard error of measurement (SEM), and the minimal detectable change (MDC) for each of the outcome measures are listed in Table 1. If the MDC was not provided by the researchers, it was calculated using the formula $\mathrm{MDC}=1.96\times \mathrm{SEM}\;\times \sqrt{2}$ [38].

### 2.4. Trainees’ Satisfaction towards Intervention

The trainees’ satisfaction following completion of each intervention was assessed by implementing the Felling Scale [43], which examined the response of a trainee to a particular type of exercise; that is, whether an exercise was pleasant or unpleasant based on a 10-point scale that ranged between −5 and +5, with −1 to −5 ranging from fairly bad to very bad, 0 being neutral, and +1 to +5 ranging from feeling fairly good to very good.

### 2.5. Statistical Analysis

All the data were checked for normality with the Shapiro–Wilk test followed by visual inspection of the Q-Q plot and the box plot. To achieve statistical significance with a = 0.05, 80% power, and effect size (f) = 0.2526 (calculated based on a partial η^2^ = 0.06), an a priori power analysis using an online statistical power analysis application (G*Power v. 3.0.10; https://www.hhu.de accessed on 22 November 2021) was performed to determine an adequate sample size. The results of the power analysis indicated a total sample size of 23 subjects, which approximated the number of the participants that took part in the current study.

After confirming the normality of the data, statistical analysis regarding the differences between interventions before and after their implementation was examined using two-way ANOVA for repeated measures. Sphericity was determined based on the Mauchly’s Test, and the Greenhouse–Geisser correction was used when sphericity was significant. Significant main effects were followed by pairwise comparisons after controlling for type I errors using a Bonferroni adjustment.

The effect size was determined based on Cohen’s *d* using the formula Cohen’s *d =*
$\frac{M_{1}-{\;M}_{2}}{{\mathrm{sample}\;\mathrm{SD}}_{\mathrm{pooled}}}$, where sample
$SD_{\mathrm{pooled}}=\sqrt{\frac{(n_{1}-1)\times SD_{1}{}^{2}+(n_{2}-1)\times SD_{2}{}^{2}}{\left(n_{1}+n_{2}-2\right)}}$
; $M_{1},M_{2}$ and ${\mathrm{SD}}_{1}$, ${\mathrm{SD}}_{2}$ are the means and the standard deviations of the means for the pre- and post-intervention measurements; and $n_{1},\;n_{2}$ are the sample sizes in the two conditions. A Cohen’s *d* equal to 0.2 was considered a “small” effect size, 0.5 represented a “medium” effect size, and 0.8 a “large” effect size [44]. Statistical analyses were conducted in SPSS, version 26.0 (IBM Corp, Armonk, NY, USA), and the level of significance was set at *p* < 0.05.

## 3. Results

### 3.1. Range of Trunk Movements

Our findings revealed significant time main effects for the ROM of trunk movements such that the ROM of T-HF (F = 48.674, *p* < 0.001), TSFR (F = 15.524, *p* < 0.001), and TSFL (F = 31.002, *p* < 0.001), as well as TRR (F = 34.346, *p* < 0.001) and TRL (F = 50.961, *p* < 0.001), was increased using either SSSEs or FRSM. The main intervention effects for the ROM of all the trunk movements measured were non-significant.

Statistical analysis also yielded a significant intervention-by-time interaction for the ROM of TSFR (F = 4.558, *p* < 0.05) and TRL (F = 7.422, *p* < 0.05), primarily due to a significant greater increase in the ROM following FRSM (*p* < 0.001) compared to SSSEs (*p* < 0.05). The intervention-by-time interaction for the ROM of T-HF, TSFL, and TRR were not significant (Table 2).

The differences between the changes obtained after FRSM compared to SSSEs in the ROM of trunk movements were significant only for TSFR (*p* < 0.05) and TRL (*p* < 0.01).

### 3.2. Isometric Maximum Strength and Endurance of Trunk Extensors

Significant time main effects revealed for both TESmax (F = 4.282, *p* < 0.05,) and TESend (F = 5.407, *p* < 0.05). However, the intervention main effects were significant only for TESend (F = 9.598, *p* < 0.01).

A significant intervention-by-time interaction was found for both TESmax (F = 33.612, *p* < 0.001) and TESend (F = 41.825, *p* < 0.001) of trunk extensors. Pairwise post hoc comparisons revealed that trunk extensor strength and endurance were significantly increased following FRSM by 43.8 ± 27.4 N for TESmax (from 438.7 ± 191.4 N to 482.5 ± 200.2 N; *p* < 0.001) and by 26.1 ± 19.6 s for TESend (from 130.2 ± 57.6 s to 156.3 ± 65.8 s; *p* < 0.001). In contrast, SSSEs resulted in a significant decrease in TESmax by 27.4 ± 43.8 N (from 457.6 ± 203.8 N to 430.2 ± 195.3 N; *p* < 0.01) and TESend by 15.5 ± 19.8 s (from 135.7 ± 62.9 s to 120.3 ± 54.5 s; *p* < 0.001). The differences between the two interventions were significant for the TESmax (*p* < 0.05) and TESend (*p* < 0.001) of the trunk extensors only in post-intervention measurements (Figure 5 and Figure 6).

Cohen’s *d* calculations for pre- and post-intervention differences revealed small to medium effect sizes regarding Tmax and TESend for both FRSM (0.22 and 0.42) and SSSEs (0.14 and 0.26).

### 3.3. Trainees’ Satisfaction towards Intervention

Based on the Feeling Scale, the response of the participants regarding the satisfaction they expressed after the SSSE intervention was between good and very good (4.1 ± 0.9 units), and did not differ significantly from the satisfaction they expressed after the FRSM intervention (4.2 ± 0.9 units).

## 4. Discussion

The findings of the present study revealed that a single bout of SSSEs and FRSM increased the range of trunk movements, with the former protocol in general presenting smaller improvements compared to the latter. Changes in the ROM were marginally or well beyond the MDC in all movements except for TRR and TRL. The medium effect sizes calculated for most of the changes found in ROMs indicated that these changes were clinically significant. Several studies have shown an acute increase in the range of joint movements following SSSEs, particularly in the joints of the lower limbs, with changes varying between small and large depending on the duration and intensity of the exercise, the position taken by the trainee for stretching, and the population that performed the stretches [1]. On the other hand, only a small number of studies have shown an increase in the range of trunk movements after stretching the posterior trunk muscles [20,21,34]. Zakas et al. [20,21] revealed an approximately 4.4% (≈8.0°) and 7.5% (≈10°) increase in the ROM of trunk flexion in adolescent soccer players and elderly women, respectively, following three 15 s or 60 s stretching periods with a 10 s rest between sets. In an earlier study, Dvořák et al. [34] also found an increase in the ROM of trunk side flexion (6.7%, ≈4.3°) and axial rotation (2.9%, ≈2.5°) following five repetitions of stretching exercises of the respective muscles, with only the former ROM having changed significantly.

Foam rolling, on the other hand, while increasing the lower limb joints’ ROM by approximately 5% when an individual performed one to five rolls of 5 to 60 s each on the muscles of the lower limbs [45], did not show a similar effect on the trunk movements following rolling on the trunk muscles [23]. Griefahn et al. [23] demonstrated that rolling out the gluteus maximus, the erector spinae of the lumbar and thoracic spine, as well as the latissimus dorsi with the body weight for three intervals of 30 s each using a foam roller had no effect on the ROM of lumbar flexion, although thoracolumbar fascia mobility was increased by 56.5% (≈1.8 mm). The differences between the current and the previous studies most likely were related to the intervention protocols, as well as the instruments and the testing procedures used for measuring the ranges of the trunk movements. Despite the limited information on the intervention stretching protocols provided by previous studies, it seems that most investigators attempted to stretch or roll out primarily the extensor muscles of the trunk [20,21], including, in some cases, more distally located muscles such as the gluteus maximus [23]. Furthermore, some authors, focusing mainly on the lumbar spine, measured only the ROM of lumbar flexion either in the standing position with a flexometer, which is a gravity-type device [20,21], or in the sitting position with the Modified Schober test using a standard tape measure [23]. In one study, the investigators measured the ROM of the side flexion and axial rotation of the lumbar spine using a potentiometer-based 3D motion analyzer mounted on the subjects’ sacrum and thoracolumbar junction [34]. Both the SSSE and FRSM intervention protocols in the present study included stretches and rolls of the muscles of the entire torso, resulting effectively in a more apparent change in the range of trunk movements. The SSSE protocol included exercises that were meant to elongate the extensors, the side flexors, and the axial rotators of the trunk, but subsequently, these multidirectional movements may equally effectively stretch the surrounding extramuscular fascia [46,47]. The FRSM protocol also required the execution of multiple cephalocaudal rolls over the skin, fascia, and underlying muscles of the central and lateral aspects of the trunk. Furthermore, the changes in the ROM that occurred with both interventions may have been more easily identified by placing the inertia measuring device on the upper thoracic spine, thus ensuring the ROM measurements of the entire thoracolumbar spine.

Acute increases in joints’ ROM following a single bout of static stretching exercises have been attributed to the increase of stretch tolerance, which is an increased capacity of tolerating the sense of discomfort associated with stretching of tight muscles [48] and/or the decrease of the affected limb musculotendinous unit stiffness [49]. It has also been reported that the mechanical stimuli that are provided by stretching exercises may alter fascial hydration, decreasing the connective tissue stiffness [50]. These mechanisms result in temporary elastic changes in the muscle length that may sustain for a short period of time. The increase in pain threshold with a subsequent increase in stretch tolerance, such as that reported after stretching exercises, has also been found to follow a foam-rolling session, and therefore can be considered as a possible mechanism for increasing the range of trunk movements [51]. Furthermore, the increased ROMs of the trunk following FRSM may be attributed to thixotropy, a phenomenon that occurs when the viscosity of a thick fluid, such as the intracellular and extracellular fluids, is reduced or becomes more fluidlike when agitated, sheared, or stressed [52], eventually providing less resistance to movement. Such changes are supposed to be caused by the friction-induced increased temperature of the skin, fascia, and muscle tissue, as well as the shear stress generated by the direct and sweeping pressure that is exerted by rolling the muscles with a foam roller. The stimulation of myofascial proprioceptors, the reduced stiffness of the fascia, and the enhanced vascular function of both muscle and fascia are some other responses to foam roller self-massaging that may contribute to the increase in the range of trunk movements.

A factor that may have contributed to a greater increase in the range of trunk movements after FRSM compared to SSSEs most likely was related to the pressure-induced reduction in pain and stiffness produced by the latent trigger points (LTrPs) of the trunk musculature. The LTrPs are thought to be associated with muscle stiffness and restricted joint range of ROM, and unlike the active trigger points in which they eventually evolve, they are not painful unless directly compressed [53]. The prevalence of LTrPs was quite high among non-symptomatic subjects in the posterior aspect of the trunk, particularly in the upper half [54]. Although the participants in the present study were not screened for pain threshold and muscle stiffness, some of them experienced mild discomfort caused by FRSM. Based on previous findings showing an immediate reduction in the pressure pain threshold after using a foam roller on the gastrocnemius and the plantar fascia with a concomitant increase in the ROM of the ankle [46], it is possible that a similar response eventually resulted in an increased ROM of the trunk movements. Furthermore, the increase in range of motion could be due, in part, to the mobilization of the thoracic and lumbar vertebrae that possibly were produced as a result of the pressure exerted by the trainee’s body weight during rolling along the spine. Some researchers have shown that the reduction in joint stiffness that follows mobilization of the spine in the posterior-to-anterior direction can cause an immediate improvement in the range of motion of the lumbar spine, particularly in patients with lower back pain [55,56]. Considering the greater force exerted by the participants’ body weight compared to the force that normally applied manually by a therapist during the execution of these movements, it is possible that some degree of mobilization was induced in the underlying vertebrae, increasing the range of trunk movements.

The execution of SSSEs in the present study resulted in a significant decrease of the isometric TESmax (−6.0%) and TESend (−11.3%) of the trunk extensors, despite the implementation of short static stretching periods of the trunk muscles (30 s) to avoid the detrimental dose-response effects of the longer static stretching periods. The strength reduction demonstrated by the trunk extensors may have been due to the isometric type of contraction and the prone position used for muscle strength testing. It has been reported that muscle strength following a static stretching exercise program exhibits a greater reduction when measured isometrically (−6.3%) as opposed to the concentric (−4.4%) or eccentric measurements of muscle strength (−4.2%), and that the decrease of the obtained isometric strength is stretch-duration-dependent [1]. A lower reduction in isometric strength should be expected when a muscle is stretched for periods of <60 s (4.5%) as oppose to stretch periods that last ≥60 s (6.8%) [1]. Furthermore, the trunk extensors strength may have appeared reduced because testing was performed in the prone position, in which the length of trunk muscles was relatively shorter [57]. This has been supported by several studies that have shown a greater strength reduction when muscles were tested at a shorter length (–10.2%), as opposed to the moderate strength gains that were presented when the strength was tested at the longest muscle lengths (+2.2%) [58,59]. Changes in tendon stiffness and the force–length relationship, the stretch-induced contractile “fatigue” or damage, the diminished electromechanical coupling, and/or the reduced central (efferent) drive are among the mechanisms that potentially affect muscle strength production following static stretching exercises [1]. There are many studies that supported the contribution of these mechanisms in reducing muscle strength, while others questioned them. However, further analysis of these mechanisms is beyond the scope of the present study, and readers are encouraged to study them more thoroughly in published articles.

In contrast to the SSSEs, the post-rolling strength assessment revealed increased isometric TESmax and TESend of the trunk extensor muscles by 10% and 20%, respectively, confirming previously reported findings [16,17] to some extent. The obtained changes were far greater compared to those reported for the strength of the lower extremity muscles, where the changes were considered, in general, small to negligible (+1.8%) [45]. The increased strength of the trunk extensors to a point that surpassed the increase in the strength of the lower limb muscles following FRSM by 10-fold may be partially explained by the different physiological properties of the erector spinae, the strongest extensor of the trunk. The erector spinae is a muscle that has a large percentage of type I fibers (slow twitch), which confirms the functional role it plays in opposing gravity and controlling forward inclination of the trunk, with its diameter being larger than those located in the muscles of the extremities [60,61]. As a result of this, the potential for force production of the muscle following FRSM was greater compared to the muscles of the lower extremities. The single bout of FRSM may also have induced a more effective circulatory response to the muscle [62,63], as the erector spinae demonstrates high vascularization [64], a property that makes it more suited to activities that require high levels of muscular endurance. The high blood flow that was still present even after 30 min of performing foam rolling [62] may have enabled the muscle to work optimally even at the end of the experimental procedure, when the isometric testing of muscle strength and muscular endurance was performed. The reduction in stiffness and the subsequent relaxation of the thoracolumbar fascia caused by the FRSM [23,46] could also positively affect the stability of the spine, thus enabling the extensor muscles of the trunk to act more efficiently. This could be justified by alterations in the mechanosensation in the lumbar area, and possibly the thoracic area, induced by the contraction forces generated by the thoracolumbar fascia. The thoracolumbar fascia, a structure with significantly higher density in myofibroblasts (specialized connective tissue cells with augmented contractile properties) compared to other fasciae located in the extremities, such as the human plantar fascia or the human fascia lata [65], can exert contraction forces in the lumbar region that have been mathematically predicted to range between 0.95 and 2.63 N. These forces, although unlikely to have a direct impact on the mechanical stability of the lumbar region [66], are well above the much lower threshold for mechanosensory stimulation in the lumbar area [67], potentially affecting the neuromuscular coordination and reflex regulation of functional joint stability in the area [68]. Nevertheless, the findings regarding the strength changes following the FMRM and SSSEs should be viewed with caution, as only the TESmax after the rolling session was greater than an MDC. Moreover, the effect sizes were small for most changes found in the strength of the trunk extensors after implementation of both relaxation protocols, suggesting that the clinical importance of our findings was limited.

Our findings did not confirm the superiority of the SSSEs against the FRSM, or vice versa, regarding the satisfaction of the trainees with the intervention programs. The same level of satisfaction between the two interventions was most likely due to the submaximal tension and pressure exerted during their execution, and consequently to the limited but similar discomfort experienced by the participants. After all, participants had to feel comfortable performing the interventions, and as far as the less-familiar FRSM was concerned, only those who could tolerate the pressure from the roll were allowed to participate. Based on these findings, it is reasonable to assume that both interventions can be used alternatively, provided that the discomfort induced, if any, can be tolerated by the trainees.

The results of this study should be evaluated in the light of some limitations that may impede their generalization to other populations. These limitations included the characteristics of the study participants and the performance-related elements of the intervention programs. The selection of the participants was based on: (i) the fact that more flexible individuals are not necessarily less susceptible to stretch-induced changes than untrained ones [2]; and (ii) the lack of differences in terms of stretching response between genders, although females in general are more flexible than males [69]; hence healthy males and females with no apparent musculoskeletal stiffness or pain and a variety of trunk ROMs were included in the study sample. However, the effects of these interventions were expected to differ in individuals with musculoskeletal disorders such as lower back pain. In addition, both SSSEs and FRSM were performed with intensity ‘‘just before’’ the point of discomfort (POD), as previous findings showed that stretching with an intensity of less than the POD increased flexibility more than stretching at the POD without affecting muscle performance [32]. Furthermore, FRSM was performed using a foam roller with a smooth surface and moderate hardness, as opposed to a multidensity foam roller that is supposed to increase the joint ROM more [70,71]. Changes in the range of trunk movements and the strength of the trunk extensors could be different if the tensile strength of the musculoskeletal unit or the pressure applied to the underlying tissues were increased by performing static self-stretching exercises with greater intensity or using a foam roller with different layers of hardness or surface pattern, respectively. Finally, it should be noted that this was a single-blind study, since only the subjects were unaware of the outcome of the intervention programs. Although verbal encouragement was not given during either the monitoring of the intervention programs or the evaluation of the participants to minimize the potential bias of the researcher, this cannot be ruled out.

## 5. Conclusions

Based on the findings of the present study, a single 7-min SSSE session and a single 7-min FRSM session were equally effective in acutely increasing the range of trunk movements. Despite the limited clinical significance, the increase in trunk extensor strength that was obtained following FRSM gave an additional advantage to this intervention over SSSEs, the implementation of which induced a decrease in the strength of these muscles. In the light of this information, healthcare providers can recommend the use of a foam roller as an alternative relaxant means to maintain or increase the mobility of the trunk and the strength of the trunk extensor muscles, particularly in situations in which the required performance must be optimal, such as in manual work and sports activities.

## Figures and Tables

**Figure 1 sports-09-00159-f001:**
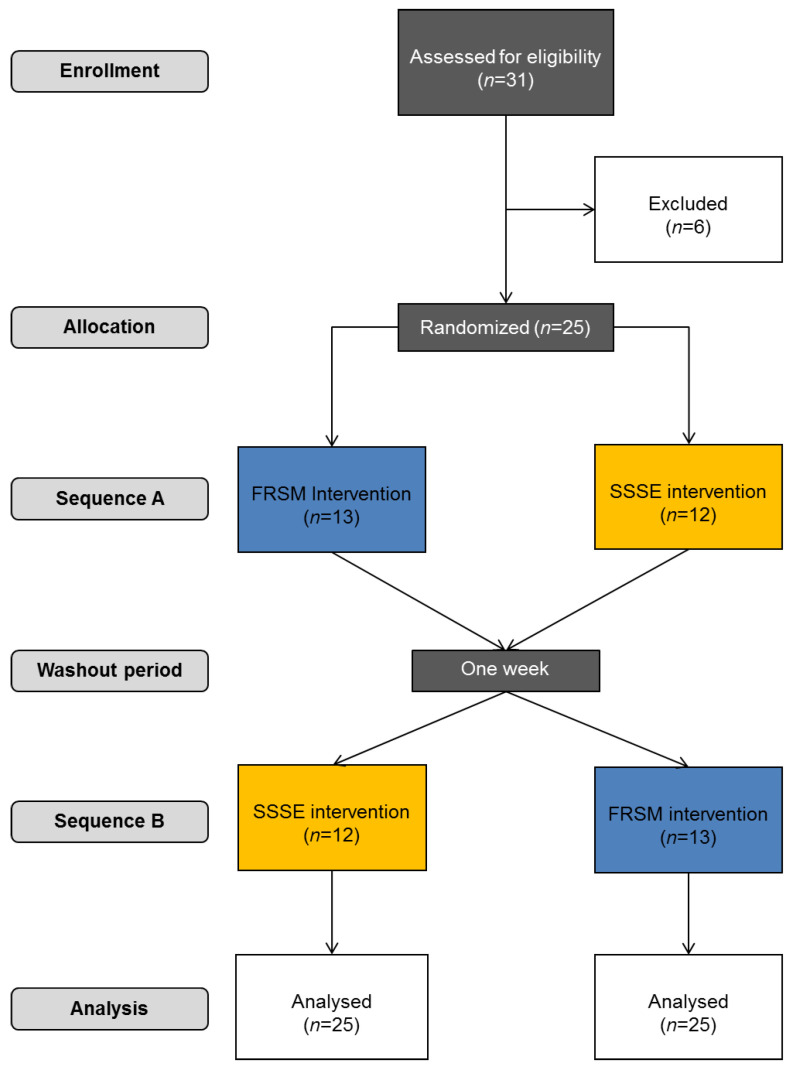
CONSORT flowchart.

**Figure 2 sports-09-00159-f002:**
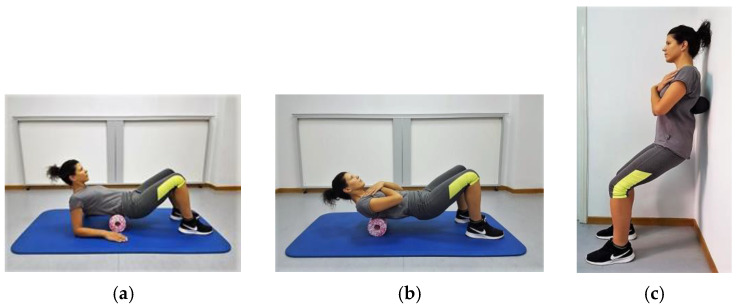
(**a**) Cephalocaudal rolls on the lumbar spine; (**b**) on the thoracic spine over the midline, the right, and the left side of the trunk with a cylindrical foam roller; and (**c**) on the thoracolumbar spine with a peanut-type foam roller.

**Figure 3 sports-09-00159-f003:**
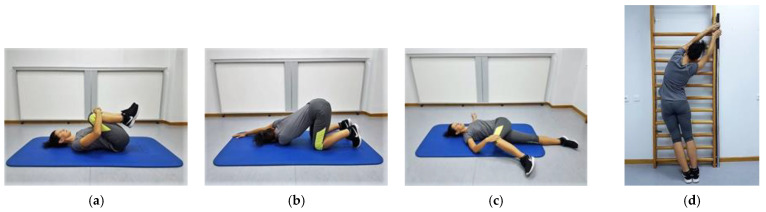
Stretching exercises for the: (**a**) trunk extensors; (**b**) rotators of the upper part of the trunk; (**c**) rotators of the lower part of the trunk towards the right and left side; (**d**) side flexors of the trunk towards the right and left side.

**Figure 4 sports-09-00159-f004:**
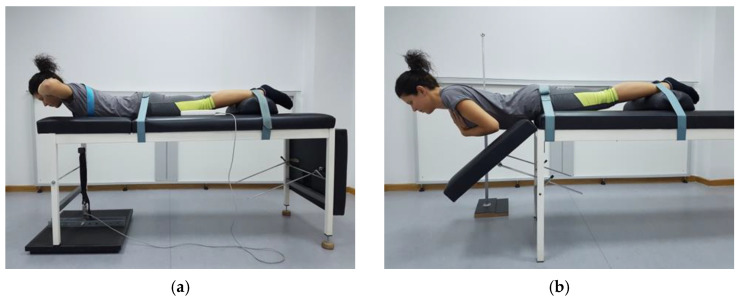
(**a**) Experimental setup for testing the isometric maximum strength of the trunk extensors; (**b**) experimental setup for testing the isometric endurance strength of the trunk extensors.

**Figure 5 sports-09-00159-f005:**
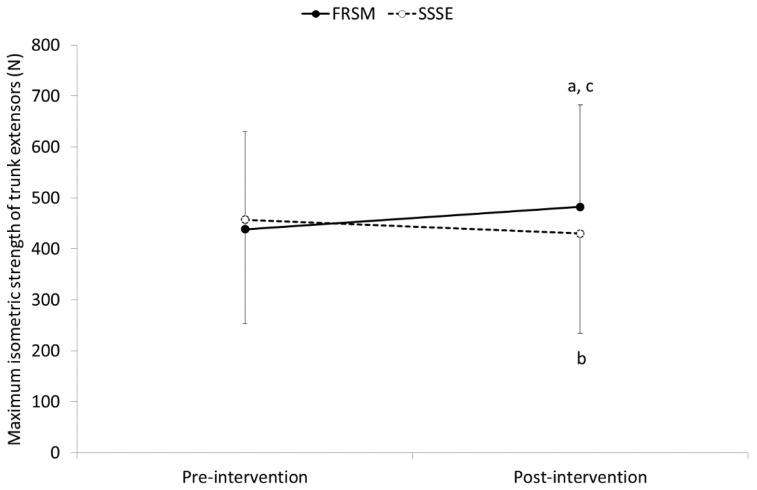
Means and standard deviations (error bars) for maximum isometric strength of the trunk extensors before and after implementation of foam roller self-massaging (FRSM) and static self-stretching exercises (SSSEs). ^a^ *p* < 0.001 significantly greater compared to pre-FRSM TESmax measurements; ^b^ *p* < 0.01 significantly lower compared to pre-SSSEs TESmax measurements; ^c^ *p* < 0.05 significantly different compared to post-SSSEs at postintervention measurements.

**Figure 6 sports-09-00159-f006:**
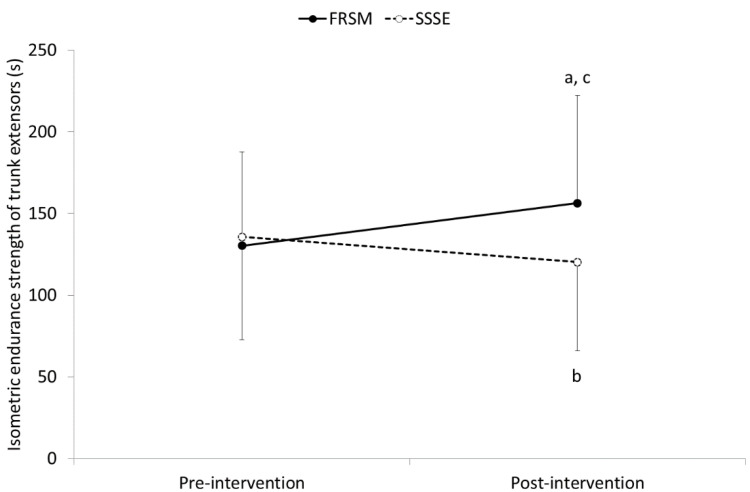
Means and standard deviations (error bars) for isometric endurance strength of the trunk extensors before and after implementation of foam roller self-massaging (FRSM) and static self-stretching exercises (SSSEs); ^a^ *p* < 0.001 significantly greater compared to pre-FRSM TESend measurements; ^b^ *p* < 0.001 significantly lower compared to pre-SSSEs TESend measurements; ^c^ *p* < 0.001 significantly different compared to post-SSSEs at post-intervention measurements.

**Table 1 sports-09-00159-t001:** Infraclass correlation coefficient (ICC), standard error of measurement (SEM), and minimal detectable change (MDC) of each outcome measure.

Outcome Measure	ICC	SEM	MDC
T-HF (cm) [39]	0.92	0.3	1.4
TSFR (°) [40]	0.92	0.8	2.2
TSFL (°) [40]	0.98	0.6	1.7
TRR (°) [40]	0.96	3.2	8.9
TRL (°) [40]	0.99	2.1	5.8
TESmax (N) [41]	0.93	13.0	36.0
TESend (s) [42]	0.86	15.2	42.1

**Table 2 sports-09-00159-t002:** Means ± standard deviations, pre- and post-intervention changes (diff), and effect sizes (Cohen’s *d*) of trunk ranges of motions before and after implementation of foam roller self-massaging (FRSM) and static self-stretching exercises (SSSEs).

	Foam Roller Self-Massaging				Static Self-Stretching Exercises			
	Pre-FRSM	Post-FRSM	Diff	Cohen’s *d*	Pre-SSSEs	Post-SSSEs	Diff	Cohen’s *d*
T-HF (cm)	36.9 ± 8.2	40.6 ± 8.3 ^b^	3.7 ± 3.3	0.45	35.8 ± 9.0	39.6 ± 8.1 ^b^	3.8 ± 3.6	0.44
TSFR (°)	50.2 ± 9.9	54.1 ± 10.4 ^b^	3.9 ± 4.4	0.39	50.9 ± 9.8	53.0 ± 8.8 ^a^	2.1 ± 4.3 ^c^	0.22
TSFL (°)	51.6 ± 9.7	55.5 ± 10.7 ^b^	3.9 ± 4.5	0.38	51.5 ± 8.4	55.6 ± 10.7 ^b^	4.0 ± 4.1	0.42
TRR (°)	55.9 ± 8.6	60.6 ± 9.3 ^b^	4.8 ± 5.0	0.53	57.2 ± 7.5	60.8 ± 8.3 ^b^	3.6 ± 4.5	0.46
TRL (°)	56.3 ± 9.1	63.0 ± 8.2 ^b^	6.4 ± 4.1	0.74	58.5 ± 8.1	61.3 ± 8.4 ^a^	2.8 ± 5.1 ^d^	0.34

T-HF: trunk-hip flexion; TSFR: trunk side flexion right; TSFL trunk side flexion left; TRR: trunk rotation right; TRL: trunk rotation left; ^a^ *p* < 0.05; ^b^ *p* < 0.001 significantly greater compared to pre-FRSM and pre-SSSEs ROM measurements; ^c^ *p* < 0.05; ^d^ *p* < 0.01 significantly different compared to the changes obtained after FRSM.

## Data Availability

The data presented in this study are available upon request from the corresponding author.

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
