# Peer review of "Acute Effects of Static Self-Stretching Exercises and Foam Roller Self-Massaging on the Trunk Range of Motions and Strength of the Trunk Extensors"

_sports, 2021, doi:10.3390/sports9120159_

Round 1

Reviewer 1 Report

Comment to the Author

Sports

Acute Effects of Static Auto-Stretching Exercises and Foam Roller Self-Massaging on the Trunk Range of Motions and Strength of the Trunk Extensors

sports-1460725

General comment:

Thank you very much for the opportunity to review this very interesting paper on the effects of foam roller and stretching on the posterior trunk. This is a very rare field of research and I think it is very valuable. I felt that there were no major problems in the paper. However, there are a few minor points that need to be corrected. Please refer to the following to make your paper better.

  • Page3, line108

This time the intervention time is set to 7 minutes, why 7 minutes? Please explain the rationale for the setting using the literature.

  • Page Line143

Why are auto-stretching exercises 30 seconds per category? In order to increase the range of motion, I would like to take more time.

  • Page6 Line225

Was the data normally distributed this time? If you are checking for normality, please describe it.

  • Page11, line433-441

The results of this study were obtained after 7 minutes of 30-second stretching, and the results are expected to differ depending on the intervention time. Therefore, if we do not mention in the conclusion that the results were obtained with the intervention time of 7 minutes, the expression "with one intervention" may mislead the readers.

Reviewer 2 Report

Thank you very much for giving me a chance to review the interesting paper comparing foam rolling and static stretching on trunk flexibility and strength. I assume that the paper is suitable for publication after some revision in Sports.

Major Concerns:

#1: The word “self-myofascial release” is an incorrect word (Behm et al., Sports Med 2019), and please revise this word.

#2: I am not familiar with “static auto-stretching exercise”, and I assume that the definition is self-static stretching exercise. Please reconsider the definition.

#3: The subjects in this study are listed as active. How active are the subjects in this study? Also, are the participants familiar with the measurement and foam rolling intervention?

#4: If you measured the reliabilities of the outcome measurement, please add the reliabilities in the method section or results section.

#5: In the discussion section, you stated the mechanism for an increase in ROM after stretching and foam rolling interventions. Please add the following studies showing the relationship between the increase in ROM and stretch tolerance after foam rolling intervention.

Nakamura M, Onuma R, Kiyono R, Yasaka K, Sato S, Yahata K, Fukaya T, Konrad A.

The Acute and Prolonged Effects of Different Durations of Foam Rolling on Range of Motion, Muscle Stiffness, and Muscle Strength. J Sports Sci Med. 2021 Mar 1;20(1):62-68.

#6: In this study, are there sex differences in response to interventions?

Reviewer 3 Report

This study is interesting. Foam rolling is relatively applied on the trunk. Authors indicated that a single bout of FRSM has more effective as compared with SASE because of increased ROM and trunk muscle strength. However, some concerns should be addressed as follows,

  1. Suggest "auto"-stretching replaces by "self"stretching throughout the manuscript.
  2. Introduction. The rationale is not clear. 
  3. "Excessive or prolonged 78 loading or direct damage to these soft tissues is likely to cause adaptations [9] and fatigue 79 [28] that affect the function of the trunk and the adjacent joints resulting in various 80 pathological conditions [29]." can be deleted 
  4. How to randomly perform two protocols? 
  5. why choose two type of foam roller? 
  6. why the positions were not different on thoracic (having two) and lumbar region (only one)?
  7. "the amount of stretching did produce 143 minimal pain or discomfort." provide a reference. 
  8. How about the ICC for those outcome measurements? 
  9. Results. Please provide the CONSORT flowchart. 
  10. Resuls. Firstly,please clarify the time main effect, intervention main effect, and time*intervention effect (F=?, P=?), respectively. Also, provide the change difference comparison (P=?) 
  11.  Result. Please provide the minimal detectable change (MDC) value for each outcome measurements, indicating a real change after intervention. In addition the effect size ( may be cohen's d) for differences (postintervention-preintervention) is needed. 
  12.  Any adverse effect after FRSM? Suggest to clarify. Because the foam rolling on back muscle may be painful/harmful due to thoracic and lumbar spinal process. 
  13. Does this study cannot rule out movement with joint mobilization while performing foam rolling on the back muscles? That's why the ROM could be increased rather than only myofascial issues. Please explain. 

Round 2

Reviewer 3 Report

Thank you for the revision.Congratulations.